# Efficacy of Shu-yi-ning-chang decoction on IBS-D: Modulating Nr4a3 pathway to reduce visceral hypersensitivity

Yajing Guo[1], Qiongqiong Lu[2], Xiao-Jun Yang💿[1,2]*, Yuxi He[2], Yue Wu[2], Baijun Qin[2], Ting Li[3], Min Duan[4], Nvping Liu[5], Xin Wu[5], Yuanjun He[2]

1 Department of Clinical Medicine, Chengdu University of Traditional Chinese Medicine, Chengdu, Sichuan, China, 2 Department of Gastroenterology, Chongqing City Hospital of Traditional Chinese Medicine, Chongqing, China, 3 Department of Pharmaceutical, Chongqing Medical University, Chongqing, China, 4 Department of Clinical medicine, Changsha Hospital of Traditional Chinese Medicine Affiliated to Hunan University of Traditional Chinese Medicine, Changsha, Hunan, China, 5 Department of Clinical medicine, Guizhou University of Traditional Chinese Medicine, Guiyang, Guizhou, China

* yangxj88@126.com

**Data Availability Statement:** All relevant data are within the paper and its Supporting Information files.

## Abstract

### Aim of the study

To evaluate the therapeutic effect of SYNC in diarrhea irritable bowel syndrome (IBS-D) and explore its underlying mechanism through transcriptomic sequencing (RNA-Seq).

### Materials and methods

A rat model of IBS-D was constructed to elucidate the effects of SYNC. Abdominal withdrawal reflex (AWR), fecal water content (FWC), and recording body weight were calculated to assess visceral sensitivity in rats. Histopathological changes in the colon and alterations in mast cell (MC) count were determined. Immunohistochemistry was employed to assess mast cell tryptase (MCT) expression in rat colons. Serum levels of corticotropin-releasing Hormone (CRH), interleukin-6 (IL-6), calcitonin gene-related peptide (CGRP), and 5-hydroxytryptamine (5-HT) were quantified using ELISA. RNA-Seq of colon tissue was performed, followed by Gene Ontology (GO), Kyoto Encyclopedia of Genes and Genomes (KEGG) pathway enrichment analyses. Western blot analysis was conducted to quantify the expression levels of key proteins in the Nr4a3 pathway in the colon and hypothalamus tissues of rats.

### Results

SYNC alleviated visceral hypersensitivity and mood disorders in rats with IBS-D. Moreover, it was positively correlated with its dosage and the observed effects, such as the enhancement of the colon's mucosal lining condition and reduction in the number and activation of MCs within the model group. SYNC reduced the expression levels of factors related to the brain-gut axis and inflammatory markers in the bloodstream. RNA-Seq analysis indicated that SYNC down-regulated the expression of Nr4a3 and PI3K. These SYNC-targeted genes primarily played roles in immune regulation and inflammatory responses, correlating with

**Funding:** Chongqing Natural Science Foundation (No: cstc2021jcyj-msxmX0858) and National Natural Science Foundation of China (82205085).

**Competing interests:** The authors have declared that no competing interests exist.

**Abbreviations:** IBS-D, diarrhea-predominant irritable bowel syndrome; RNA-Seq, RNA sequencing; UPLC-MS/MS, ultra-performance liquid chromatography/tandem mass spectrometry; AWR, Abdominal withdrawal reflex; FWC, fecal water content; MC, mast cell; MCT, mast cell tryptase; CRH, corticotropin-releasing Hormone; IL-6, interleukin-6; CGRP, calcitonin gene-related peptide; 5-HT, 5-hydroxytryptamine; GO, Gene Ontology; KEGG, Kyoto Encyclopedia of Genes and Genomes; Nr4a3, nuclear receptor subfamily 4 group A member 3; PI3K, Phosphatidylinositol 3-kinase; PND, postnatal day; OFT, Open field test; SPT, sucrose preference test; PAR-2, Protease-activated receptor-2.

the modulation of Nr4a3 and the PI3K/AKT pathway. Western blot analysis further confirmed SYNC's influence on inflammation-related MC activation by downregulating key proteins in the Nr4a3/PI3K pathway.

## Conclusions

SYNC inhibited mast cell activation and attenuated visceral hypersensitivity in the colon tissues of IBS-D rats. These effects were mediated by the Nr4a3/PI3K signaling pathway.

## Introduction

Irritable Bowel Syndrome (IBS) is a widespread gastrointestinal disorder characterized by brain-gut interaction, impacting a substantial global population. It is commonly observed in women, with a prevalence varying from 1.1% to 45%, significantly impacting quality of life, social functioning, and imposing a high financial burden on the society. Prolonged symptomatic distress contributes to heightened mental stress among patients [1]. IBS is primarily identified by persistent or intermittent abdominal pain and bloating, associated with alterations in stool frequency or form. Moreover, it encompasses various physical, visceral, or psychological comorbidities. Surveys have revealed that up to 84% of IBS patients grapple with depression, while 44% suffer from anxiety, thereby labeling it as both a physical and mental illness [2]. This dual burden significantly impairs the quality of life of patients. Presently, the diagnosis of IBS primarily hinges on gastrointestinal symptoms, relying on the exclusion of organic diseases. It does not consider mental states and other visceral manifestations. The precise pathogenesis of IBS remains elusive. However, current understanding suggests a complex interplay among factors including gut microbiota, mucosal immune system, compromised mucosal barrier function, visceral hypersensitivity, gut motility, and alterations in the gut-brain axis [3–5]. The emergence of the "liver-brain-gut" axis, linking the central and peripheral systems of irritable bowel syndrome, offers a comprehensive framework for understanding the multifaceted nature of intestinal symptoms, abdominal pain, and psychological alterations in IBS. This multidirectional regulatory axis represents an interactive network between the central nervous system and the gastrointestinal tract, involving multiple systems, such as the nervous, endocrine, and immune systems. It operates through both top-down and bottom-up interactions to regulate mood and gastrointestinal functions [6, 7]. Stress and adverse emotions activate the brain-gut-related bio-axis, and the CRH system can activate CRH-R1 thereby enhancing IgE-mediated allergic responses and triggering MCs degranulation in response to psychological stress [8]. The brain transmits various neurological and endocrine signals to the gut, impacting MCs activity, neurotransmission within the autonomic nervous system, and alterations in gut barrier function, all of which are associated with the pathogenesis of IBS [9–11].

Concerning treatment, dietary interventions are prioritized, with drug therapy primarily recommended for symptom management. This approach is often supplemented by the administration of antidepressants and antispasmodics. Nevertheless, these interventions mainly provide only temporary relief, characterized by limitations and a high recurrence rate. Consequently, there is a pressing need to explore novel treatment modalities and pharmaceutical options for IBS. Traditional Chinese medicine (TCM) has been used to treat gastrointestinal diseases. It focuses on the liver and spleen function, aiming to disperse stagnant liver qi and strengthen the spleen. Through this mechanism, it improves communication between the brain and intestines, coordination between upper and lower systems, emotional well-being,

and normal qi movement. Overall, TCM provides a holistic approach to balance the bowels and relieve abdominal pain [12–14]. The above TCM theories coincide with modern medicine's perspective of the brain-gut interaction. Both ChaiHu Shugan San and Si Jun Zi decoction are classic TCM formulas, which were incorporated into SYNC. They comprise the following drugs: *Radix bupleuri* (Chai Hu) 15g, *Angelica sinensis radix* (Dang Gui) 10g, *Fructus Aurantii* (Zhi Qiao) 10g, *Radix Paeoniae Alba* (Bai Shao) 10g, *Rhizoma Corydalis*(Yuan Hu) 10g, *Radix Pseudostellariae* (Tai Zi Shen) 15g, *Poria* (Fu Ling) 10g, *Rhizoma Atractylodis Macrocephalae* (Bai Zhu) 10g, *Dried tangerine peel* (Chen Pi) 10g, *Pericarpium Granati* (Shi Liu Pi) 10g, *Bergamot* (Fo Shou) 10g, *Citron* (Xiang Yuan) 10g, with precise clinical efficacy, but its precise mechanism remains unknown. Based on current research, we focus on activation of brain-gut-related MCs. In this study, we aimed to assess the therapeutic potential of SYNC in rat models. We employed transcriptomics sequencing and molecular biology experiments to uncover the mechanisms underlying the efficity of SYNC in the treatment of IBS-D. This will provide new perspectives for the treatment of IBS-D.

## Materials and methods

### Materials

SYNC granules were purchased from China Resources Sanjiu Medical (Jiangxi, China). Pinaverium Bromide was purchased from Abbott Healthcare SAS (lot number 715532, France). Acetic acid and sucrose were purchased from McLean Biochemical Technology Ltd (lot numbers A801296 and S818046, respectively, Shanghai, China).

### Animals

A total of twelve Sprague-Dawley pregnant rats (15 days gestation, SPF grade) were purchased from Beijing Vital River Laboratory Animal Technology Co, Ltd, under license No. SCXK (Beijing) 2021–0006. These animals were housed in the SPF-grade experimental animal room at Chongqing Hospital of Traditional Chinese Medicine. They were kept under controlled conditions with a 12:12-hour light-dark cycle, a temperature of 23˚C, and humidity ranging from 40% to 60%, and were provided with free access to water and a standard diet. The number of litters for pregnant rats was recorded daily, with the day of birth defined as postnatal day 0 (PND0). After birth, the pups were nursed by their respective mothers until weaning (PND22). Subsequently, male and female pups were separated into separate cages from PND23 onwards. To eliminate hormonal influences, only offspring male rats were included in this study. The experimental animal protocol was approved by the Ethics Committee of Chongqing Hospital of Traditional Chinese Medicine (Review Opinion No. 2022-DWSY-YXJ).

### Induction of IBS-D rat model and experimental design

The IBS-D rat model was established using a combination of maternal and infant separation, acetic acid enema, and chronic restraint. The offspring male rats were randomly assigned to two groups: the control group (n = 8), which received no interventions and had a normal diet and access to water, and the model group (n = 48). In the model group, maternal and infant separation was conducted for 3 h daily (9 am-12 am) during PND1-PND21. During separation, the pups were placed in separate cages equipped with temperature-adjustable heating pads on the bottom of the cages set at 30˚C-33˚C to ensure they were not in the same room as the females. The rats received a daily injection of Acetic acid enemas from PND22 to PND31. A 0.5% acetic acid solution was introduced into the colon using a No. 8 catheter inserted 5–6

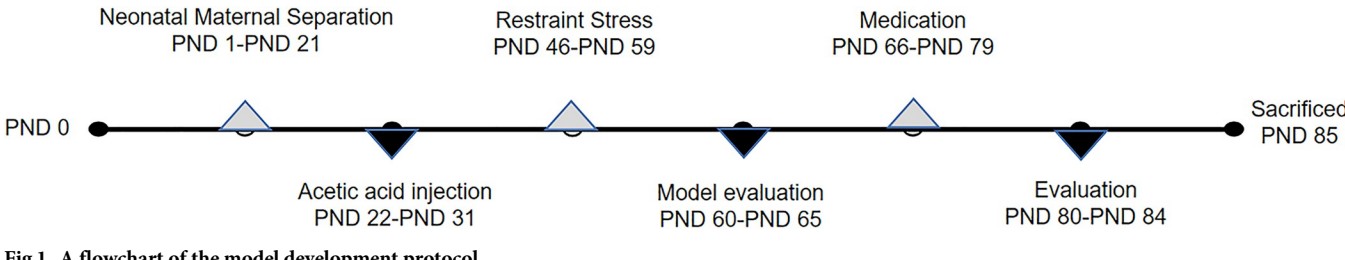

**Fig 1. A flowchart of the model development protocol.**

cm from the anus, held for 3 min to prevent leakage. The initial enema volume on the first day was 0.2 ml, progressively increasing by 0.1 ml each day until reaching a stable volume of 0.5 ml. From PND32 to PND45, rats were immobilized in an opaque container without movement for 3 h each day. At the end of the modeling period (PND46), abdominal wall withdrawal reflex scoring (AWR) and fecal water content scoring were performed to assess the success of the modeling [15, 16]. The model preparation flowchart is presented in Fig 1.

Successfully modeled rats were randomly assigned to one of five groups, the model group, SYNC low-dose (6.825 g/kg/d), medium-dose (13.65 g/kg/d), high-dose (27.3 g/kg/d), and Pinaverium Bromide group (15.75 mg/kg/d). The drug dosages were adjusted based on equivalent dosages for adults and rats and administered via gavage for 14 days. On PND65, all rats were anesthetized with sodium pentobarbital (40mg/kg, i.p) to relieve pain, and blood samples were collected from the abdominal aorta. The rats were killed through cervical dislocation, and the hypothalamus and distal colonic tissue were collected.

## Fecal water content and body weight assessment

Fecal water content (%) was determined by collecting feces from each rat between 8:00 am and 9:00 am. The wet weight of the collected feces was recorded. Next, they were dried in an oven at 120˚C for 30 min and weighed again, with this recorded as the dry weight. This process was repeated until the weight of the feces no longer changed. Fecal water content was calculated as follows: feces water content (%) = (wet weight of feces + dry weight of feces)/ wet weight of feces×100% [12]).

Additionally, rat body weights were recorded every five days, starting from the end of the modeling period (week 0) and continuing until the conclusion of the experiment, totaling six measurements.

## Open field test (OFT) and sucrose preference test (SPT)

The OFT and SPT were conducted both before and after treatment to evaluate the anxiety and depression states of the rats. These tests were performed in a quiet environment with uniform lighting. After allowing the rats to acclimate to the testing environment, they were placed in a 1m×1m×1m open-field experimental box. The rats' movements within the box were recorded for 5 min, including the number of times they entered the central zone, and surrounding zone, and engaged in upright and grooming behaviors. After each test, the experimental box was cleaned to prevent foreign objects from affecting the rat's behavior.

Before SPT, sucrose drinking training was performed. During the first 24 h, the rats were given two bottles of 1% sucrose water were provided. In the second 24 h, one of the bottles of sucrose water was replaced with regular water, and the positions of the bottles were switched halfway through the day. On the third day, the rats were deprived of food and water for 24 h. On the fourth day, they were allowed to choose between water and 1% sucrose water, and their

consumption within one hour was measured. The sugar-water bias rate for each group of rats was calculated using the formula: sugar-water bias rate (%) = sugar-water consumption / (sugar-water consumption + water consumption) × 100%.

## Abdominal withdrawal reflex (AWR) scoring

AWR scores were assigned to rats before and after treatment. Rats were fasted but provided with water 24 h before measurement. A catheter lubricated with paraffin oil was gently inserted into the rat's anus and secured to the tail with medical tape to prevent dislodgment. The rats were then placed in a transparent container, and room-temperature pure water was injected into a balloon through the catheter. Each injection lasted for 20 s, and the water volume injected was noted when the AWR score was 3. This process was repeated 3 times with a 15 min interval between injections, and the average value was used to determine the minimum volume threshold for AWR caused by non-injurious colorectal dilatation in these rats.

## Histology and immunohistochemistry

Rat colon tissues were fixed in paraformaldehyde, dehydrated step by step in ethanol, immersed in wax, embedded, sectioned into approximately 3μm slices, and subsequently subjected to HE staining toluidine blue staining, and immunohistochemistry. Hematoxylin staining was performed for 3–5 min, followed by eosin staining for 5 min, and morphological changes in colon tissue were observed under a microscope. MCs were stained with toluidine blue, and slides were initially soaked in 0.5% toluidine blue and then differentiated using acetone. Subsequently, six random fields at 400x magnification were selected for cell counting. Antigen retrieval, sealing, antibody incubation, and staining were performed on dewaxed sections. Average optical density values were calculated using the Image Pro-Plus 6.0 image analysis system.

## ELISA for the determination of CRH, IL-6, CGRP, and 5-HT in serum

Serum levels of CRH, IL-6, CGRP, and 5-HT were quantified using ELISA kits (Elabscience) following the manufacturer's instructions.

## Transcriptomic sequencing and enrichment analysis

Three colon samples were randomly selected from each of the control group, model group, SYNC group, and PI group. Total RNA was extracted using the RNAprep Pure Tissue Kit, and subsequent library construction experiments were conducted upon meeting the standard requirements for RNA sample quality. Low-quality reads containing junctions and unidentifiable reads with base counts greater than 5 were excluded from the original sequences to obtain the final data for analysis. Gene expression was determined using RPKM values calculated with featureCounts software.

For differential gene expression analysis, the read count data obtained from the gene expression level analysis were used, and DESeq2 was employed for this purpose. To mitigate the risk of false positives stemming from independent statistical hypothesis tests for a large number of genes, p-values obtained from the original hypothesis tests were adjusted. Differentially expressed genes were identified based on the criteria of $padj < 0.05$, |log2(foldchange) | > 1.

GO, a standardized functional classification system, and KEGG, a database of pathways were performed to determine the significance of differences using Fisher's exact test. The differentially expressed genes underwent functional and enrichment analyses.

## Western blot analysis

Colon and hypothalamic tissues were homogenized and sonicated on ice. Subsequently, proteins were extracted and their concentration was determined using a BCA kit. The protein samples were denatured by boiling at 100˚C and then loaded onto 10% SDS-PAGE for separation at 100 V for 1.5 h. They were then transferred to 0.22μm PVDF membranes. The membranes were blocked with a 5% skim milk powder for 1.5h and then incubated overnight with the following primary antibodies: Nr4a3 (1:1000), PI3K (1:1000), AKT (1:1000), p-PI3K (1:1000), p-AKT (1:1000) and CRH-R1 (1:1000) at 4˚C. The membranes were washed with TBST three times and incubated with secondary antibodies for 2h at room temperature. They were washed three times, for 15 min each, and developed using a electrochemiluminescence reagent. Images were analyzed and quantified using Image J software.

## Statistical analysis

Data analysis was performed using GraphPad Prism 9 software. Following a normality test and chi-square test, data conforming to a normal distribution were expressed as mean ± standard deviation ('x±s). For multiple comparisons, one-way analysis of variance (ANOVA) was used, while comparisons between two groups were conducted using a between-group t-test. A significance level of $p < 0.05$ was considered statistically significant.

# Results

## SYNC improves body weight, fecal water content, and AWR score in IBS-D rats

To evaluate the therapeutic effect of SYNC on IBS-D, the weight gain of rats in each group was recorded separately, as fecal water content, and AWR scores were measured before and after treatment. Rats in the model group exhibited significantly lower body weight and growth rate compared with those in the control group (CI [-133.5 to -70.71], $p < 0.001$). However, SYNC and PI treatments ameliorated the weight loss attributed to the modeling process (Fig 2A, $p < 0.001$). Furthermore, the fecal water content and AWR score were significantly elevated in IBS-D rats compared to the normal group (CI [12.64 to 17.86], $p < 0.001$). Post-treatment, both SYNC and PI groups exhibited reduced fecal water content (Fig 2B) and AWR scores (Fig 2C), but the difference between the PI group and the SYNC groups was not significant. HE staining of the colon tissues revealed distinct differences. In the control group, colonic mucosa appeared smooth, with intact folds and well-organized glands. No mucosal breaks, ulcerations, or edema, and pathological changes were observed. Compared with the normal group, the colonic mucosal structure in the model group remained largely intact, with widened intestinal glandular spaces. Numerous specimens exhibited substantial inflammatory cell infiltration and submucosal edema. In contrast to the model group, the colonic mucosa in the PI and SYNC-H groups demonstrated structural integrity, with glands arranged regularly, distinct layer structure, and a normal number of cup cells. Some inflammatory cell infiltration was observed. Relative to the SYNC-H group, the SYNC-L and SYNC-M groups exhibited more pronounced inflammatory cell infiltration in the colonic mucosa and wider gaps between intestinal glands (Fig 2D). These results collectively indicate that SYNC effectively alleviated symptoms associated with IBS-D in rats, with higher doses demonstrating superior efficacy.

## SYNC improves anxiety and depression in IBS-D rats

Results from the OFT indicated a significant decrease in the number of entries into both the central and peripheral zones (Fig 3A and 3B, $p < 0.001$), along with shorter distances traveled

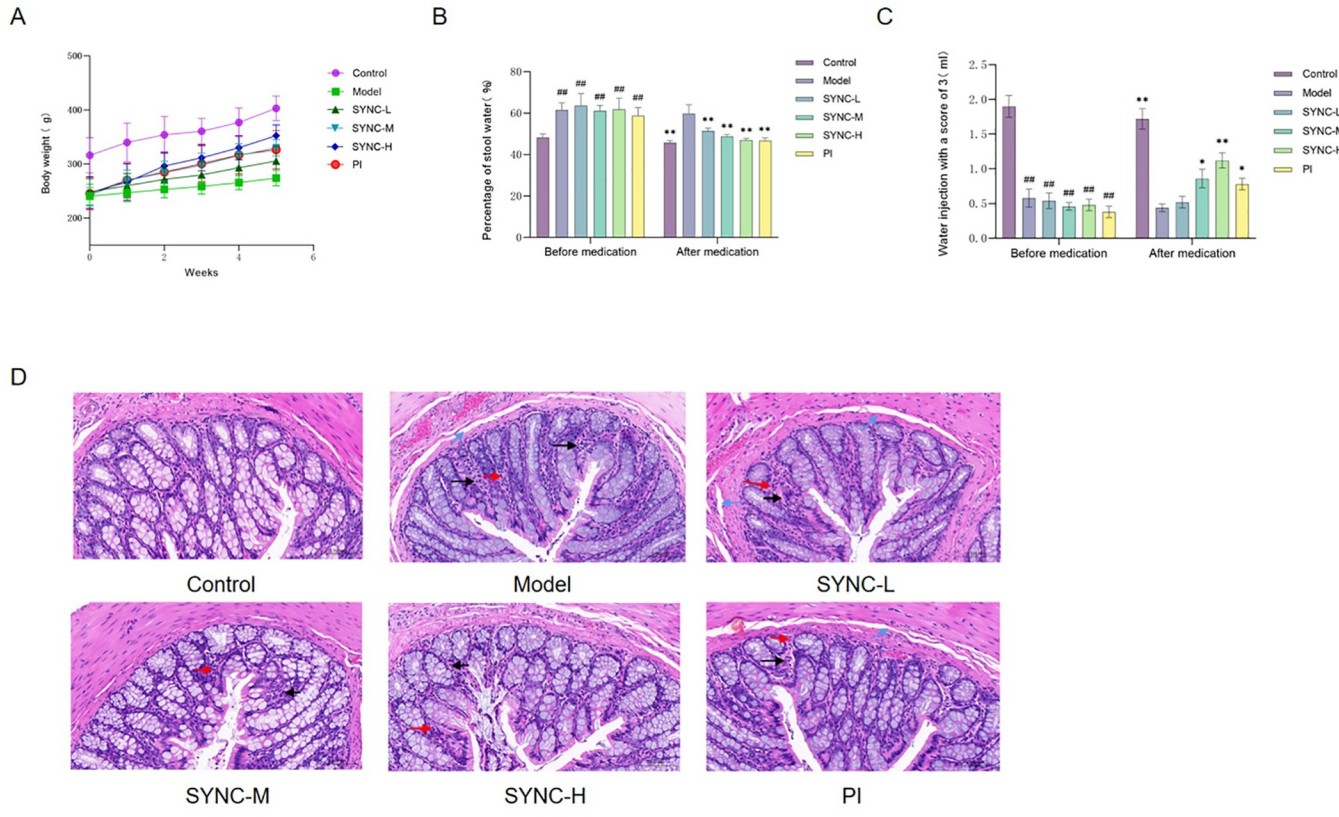

**Fig 2. Effect of SYNC on body weight, fecal water content, AWR score and colonic pathological changes in rats of all groups.** (A) Body weight (B) Fecal water content (C) AWR score and (D) HE staining results of the colon (400×). Black arrows refers to lymphocytic infiltration, red arrows indicate eosinophilic infiltration, and blue arrows represent the widening of the intestinal glandular spaces compared with the normal group, suggesting that that the fresh sample underwent several changes such as edema. One-way ANOVA was used for statistical analysis between different groups. (* $p < 0.05$, ** $p < 0.01$ vs. model group; # $p < 0.05$, # # $p < 0.01$ vs. control group, (n = 8)).

and a reduction in grooming and upright behaviors in IBS-D rats (Fig 3C and 3D). Following 14 days of SYNC treatment and PI treatment, the SYNC groups showed increased entries into central zones, groomings, and upright behaviors, with SYNC-H exhibiting the most pronounced improvement. However, no statistically significant differences were observed between the PI group and the model group in terms of peripheral zones, groomings, and upright behaviors. Combined with the OFT results, depression in rats was quantified using the sugar-water bias rate (sugar-water consumption/total fluid consumption). Compared to the control group, sugar-water consumption in the model group was significantly lower, indicating a higher propensity for depression in IBS-D rats. Post-treatment, the sugar-water bias rate significantly increased in the SYNC groups, with no statistically significant difference between the PI group and the model group (Fig 3E).

## SYNC inhibits MCs infiltration and activation in colon tissue

To explore the potential inhibitory effects of the SYNC formula on colonic MC activation, toluidine blue staining was employed to quantify the colonic MC number, and immunohistochemistry was used to assess the expression of MCT in rat colons. The results revealed a significant reduction in the number of MCs in the normal, SYNC-M, SYNC-H, and PI groups compared to the model group (Fig 4A and 4B, $p < 0.001$). Similarly, the expression of MCT

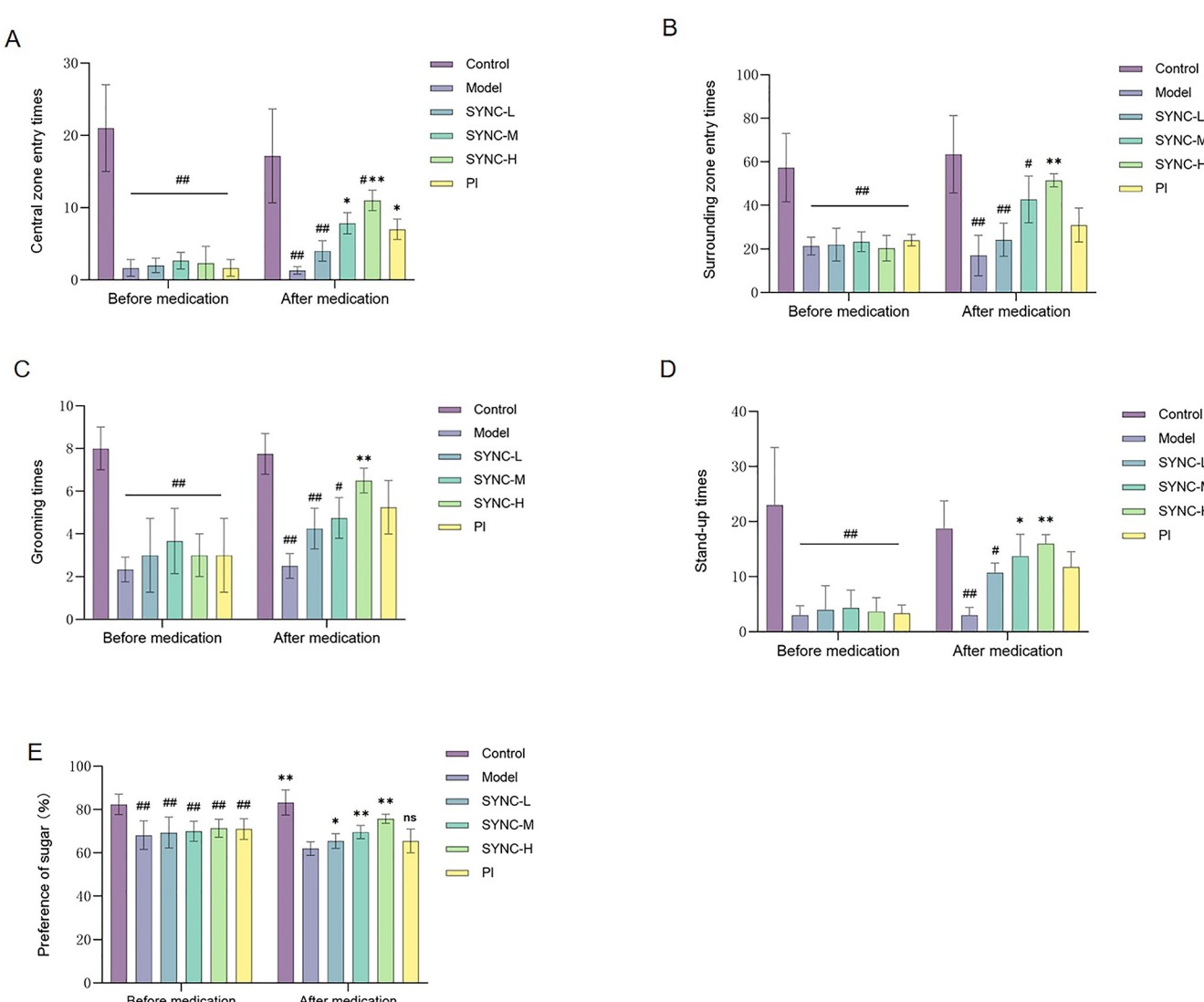

**Fig 3. SYNC reduces anxiety and depression in IBS-D rats.** The times of entries into the central (A) and surrounding zones (B) as well as the times of grooming (C) and stand-up (D) before and after treatment were recorded. The sugar-water bias rate of the SYNC group was significantly different from that of the model group (E). One-way ANOVA was used for statistical analysis between different groups. (* $p < 0.05$, ** $p < 0.01$ vs. model group; # $p < 0.05$, # # $p < 0.01$ vs. control group, n = 8).

was significantly reduced in these groups compared to the model group (Fig 4C and 4D, $p < 0.001$).

## SYNC inhibits the expression of serum IL-6, CRH, CGRP, and 5-HT in IBS-D rats

Elevated levels of serum IL-6, CRH, CGRP, and 5-HT in the model group, compared to the control group, were observed following the three-factor modeling method. However, after drug administration intervention, the expression of each index significantly decreased in the SYNC-M, SYNC-H, and PI groups compared to the model group ($p < 0.05$) as shown in Fig 5.

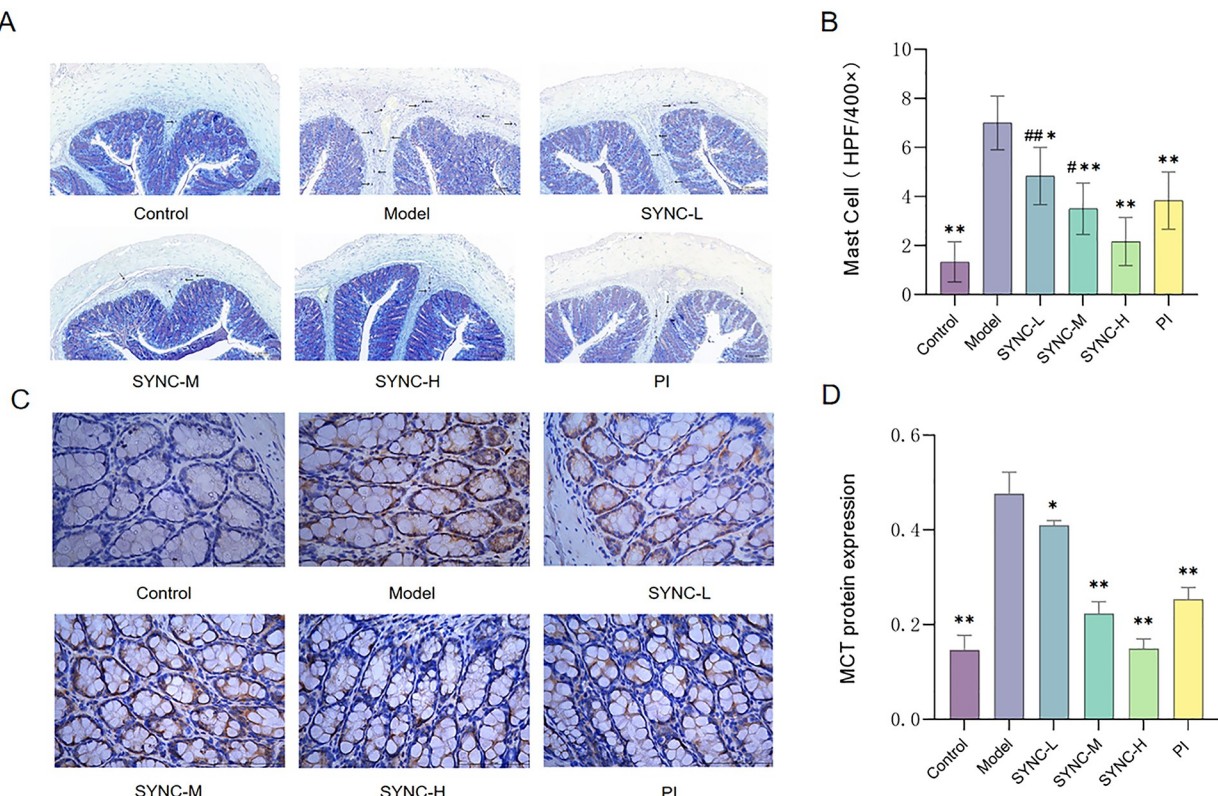

**Fig 4. Toluidine blue and immunohistochemical staining results of rat colon tissue (400x).** MCs are blue-purple after toluidine blue staining. SYNC reduced MCs infiltration and inhibited colonic MCT expression. (A-B) Number of MCs in the colon (6 randomly selected 400x magnification fields). (C-D) MCT expression in the colon. (* $p < 0.05$, ** $p < 0.01$ vs. model group; # $p < 0.05$, # # $p < 0.01$ vs. control group).

## RNA-seq analysis suggests a potential signaling pathway for SYNC treatment of IBS-D

Transcriptomic gene sequencing was performed to investigate the potential mechanism underlying SYNC treatment of IBS-D. We identified differential gene expression profiles in the colon of the control, IBS-D, SYNC, and PI groups were identified. In the model group compared to the control group, a total of 240 DGEs were identified, with 102 up-regulated and 138 down-regulated genes (Fig 6A and 6B). Notably, identified 794 DEGs were identified, with 364 up-regulated and 430 down-regulated genes in the SYNC group compared with the model group (Fig 6C and 6D). In the Pi group, 29 DEGs were identified, with 11 up-regulated and 18 down-regulated genes compared to the model group (Fig 6E and 6F). The model group, SYNC-M group, and PI group shared 2 DEGs, and 43 genes were associated with SYNC treatment, while 4 genes were associated with PI treatment (Fig 6G). These preliminary analyses suggest that SYNC does have a therapeutic effect on IBS-D. Additionally, Nr4a3 (CI [75.25 to 448.1], $p = 0.016$) and PI3K (CI [1334 to 3107], $p < 0.001$) gene expression was significantly up-regulated in the model group of rats compared with the control group, and down-regulated in the SYNC-M group (Nr4a3 (CI [-399.7 to -26.92], $p = 0.039$) and PI3K (CI [-1943 to -170.1], $p = 0.033$)). (Fig 6H and 6I). To explore the potential relationship between Nr4a3 and PI3K, STRING protein interactions network analysis was performed (Fig 6J).

GO analysis results indicated that biological processes were primarily related to immune system processes, regulation of immune responses, cellular composition focused on the

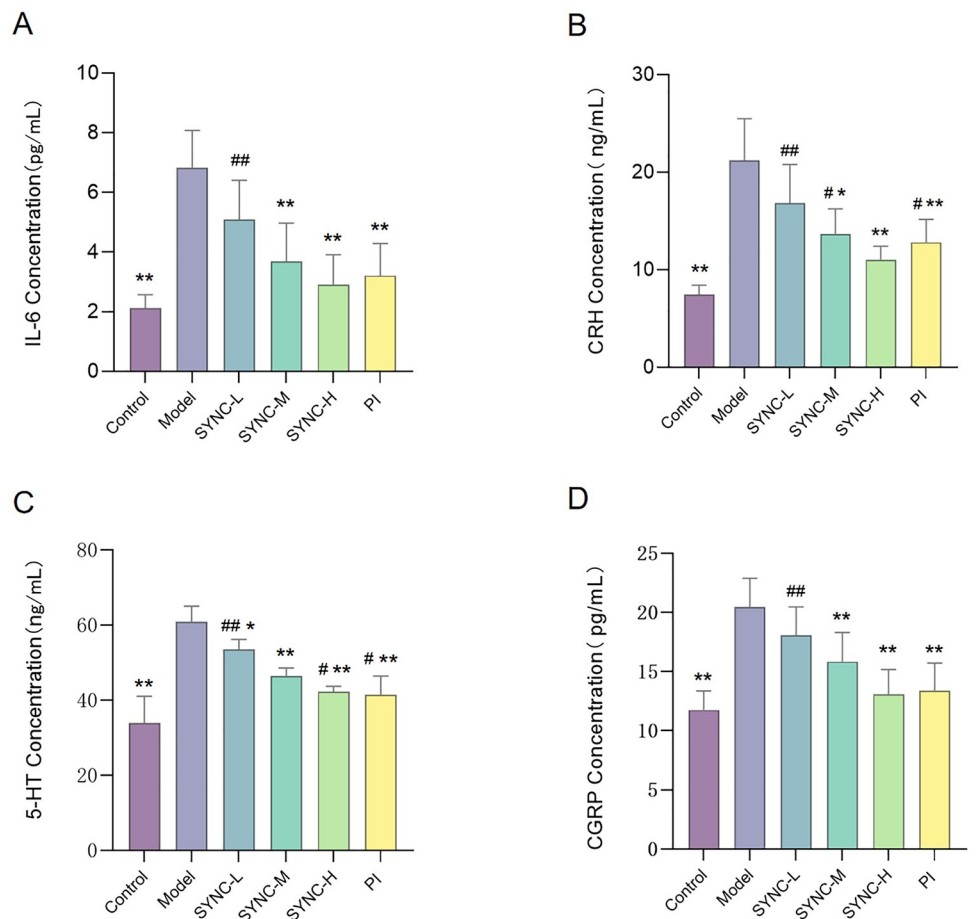

**Fig 5. SYNC treatment reduces IL-6, CRH, 5-HT and CGRP levels in IBS-D rats.** (* $p < 0.05$, ** $p < 0.01$ vs. model group; # $p < 0.05$, # # $p < 0.01$ vs. control group, n = 6).

external side of the plasma membrane, cell surface, and cell periphery, while molecular processes were mainly associated with signaling receptor binding, molecular transducer activity, and signaling receptor activity (Fig 6K). The GO enrichment bubble diagram highlights the predominant involvement of immunomodulation processes (Fig 6L). The KEGG pathway analysis, focusing on the top 20 P-values, was found to be mainly involved in intestinal immune networks associated with inflammatory immune regulation. These pathways include cell adhesion molecules, primary immunodeficiency, cytokine-cytokine receptor interactions, hematopoietic cell lines, and IgA production (Fig 6M).

## Regulation of key proteins in the colon and hypothalamus of IBS-D rats by SYNC

The effect of SYNC on Nr4a3 and related proteins in IBS-D rats was examined. In the colonic tissues of IBS-D rats, the expression levels of Nr4a3 (CI [1.755 to 4.678], $p < 0.01$), p-PI3K (CI [1.974 to 6.019], $p < 0.01$), and p-AKT (CI [1.472 to 5.088], $p < 0.01$) was higher compared to levels in the control group, while the expression of PI3K and AKT remained unchanged (Fig 7A–7F). Following SYNC treatment, there was a reduction in the expression of Nr4a3, p-PI3K, and p-AKT proteins. Moreover, SYNC treatment led to a decrease in the expression of CRH-R1 in both colon (Fig 7G and 7H) and hypothalamic tissues ($p < 0.05$) (Fig 7I and 7J).

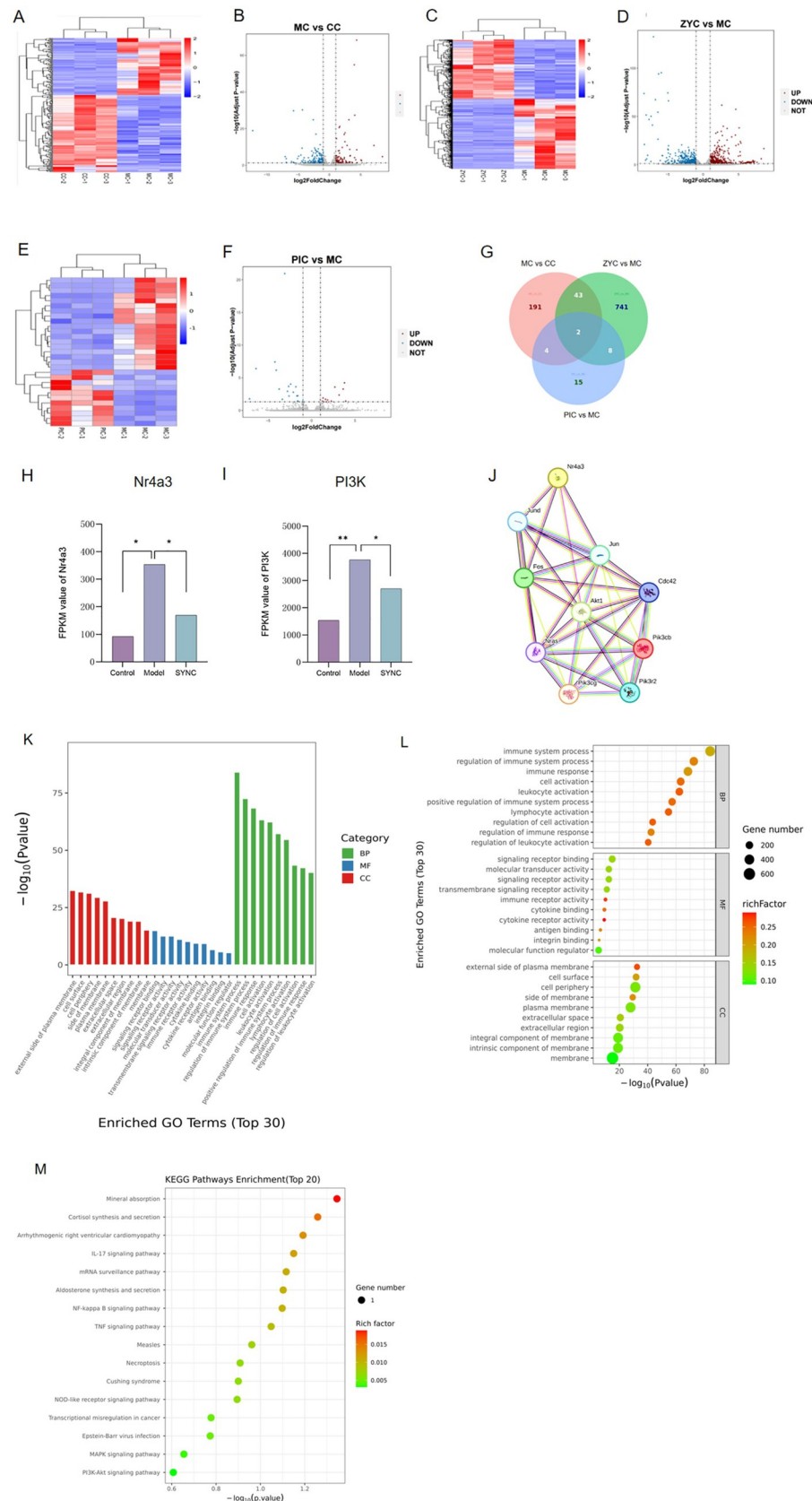

**Fig 6. Differentially expressed genes analysis, GO and KEGG pathway enrichment and enrichment and among groups.** (A–F): DEGs of rats; A and B: model vs. control; C and D: medium-dose SYNC vs. model; E and F: PI vs. model. (G): Venn diagrams. Expression of Nr4a3 (H) and PI3K (I) in individual samples and STRING protein interaction network map(J). SYNC therapeutic genes are associated with biological functions (K), enrichment bubble map (L) and differential gene KEGG pathway enrichment analysis bubble map (M).

## Discussion

Irritable bowel syndrome is a prevalent digestive system disorder driven by factors associated with societal development and lifestyle changes. Studies have demonstrated that mucosal inflammation in IBS occurs at both microscopic and molecular levels. Patients with IBS show

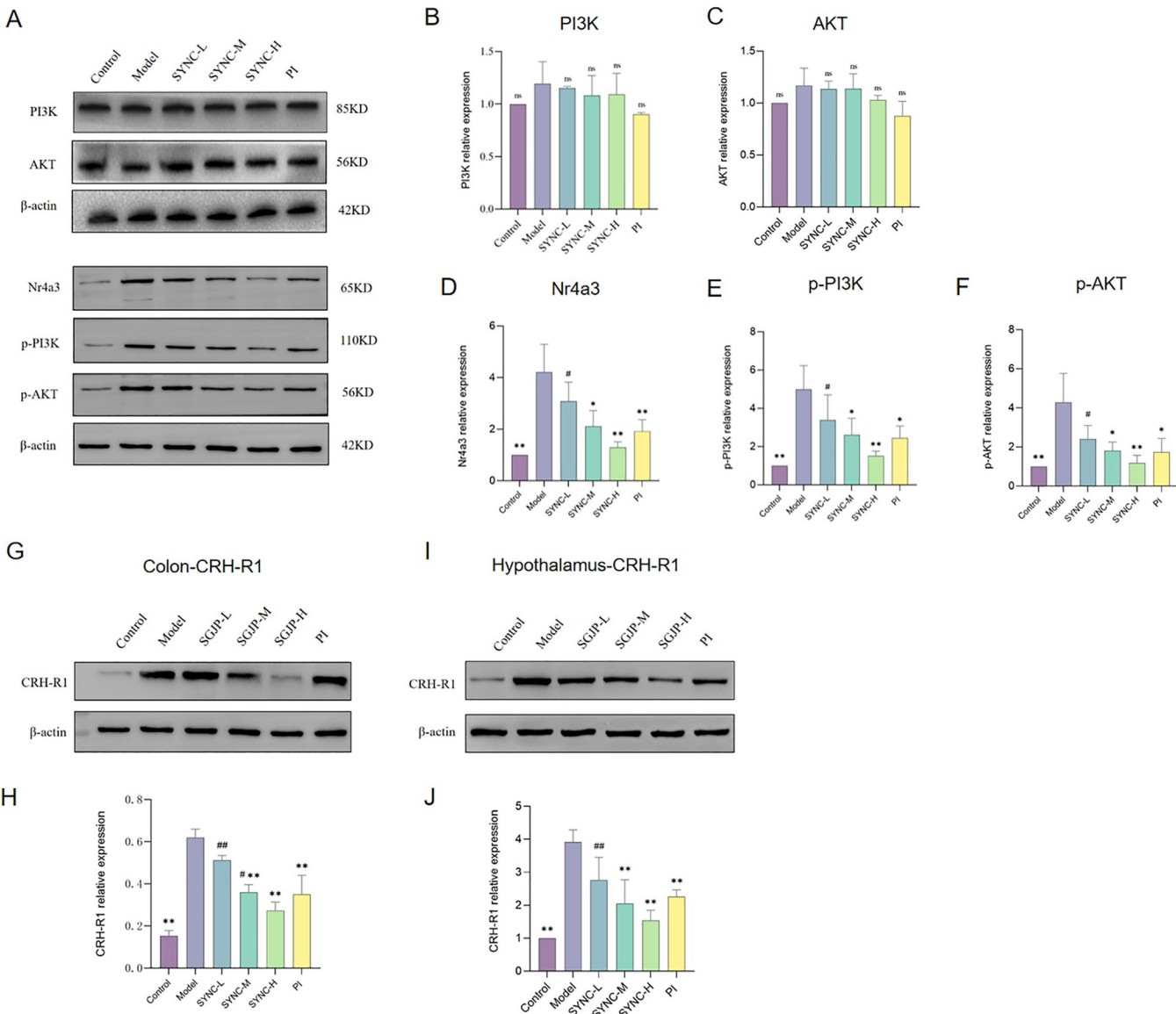

**Fig 7. SYNC inhibits colonic Nr4a3, p-PI3K, p-AKT and CRH-R1 expression, and hypothalamic CRH-R1 expression in IBS-D rats.** Expression of key proteins in the rat (A). Western blot quantification of PI3K (B), AKT (C), Nr4a3 (D), p-PI3K (E), p-AKT (F) colon CRH-R1 (H) and hypothalamus CRH-R1 (J) in the colon and hypothalamus of rats for different groups. ($*$ $p < 0.05$, $**$ $p < 0.01$ vs. model group; # $p < 0.05$, # # $p < 0.01$ vs. control group, n = 3).

increased infiltration of MCs attributed to the persistent inflammation. In addition, neuroinflammation, dysregulation of the HPA axis and 5-HT function have been reported in IBS patients [5]. Due to the incomplete understanding of IBS pathogenesis, current treatment strategies are primarily focused on symptom management. However, the high relapse rate is a notable challenge, with reports indicating that dissatisfaction with medication efficacy leads 60.1% of patients to discontinue treatment [17]. Chinese medicine, which is applied internally and externally, has been shown to improve IBS symptoms. External treatment methods, such as acupoint patches, auricular pressure seeds, and acupuncture, can significantly alleviate symptoms and improve the quality of life of patients [17–20].

Intestinal MCs regulate intestinal lumen permeability, secretion, peristalsis, innate and adaptive immunity function, among other functions. Therefore, they can influence outcomes of various gastrointestinal disorders, including organic and functional diseases [21]. MCs can modulate several neuropeptides (neuropeptide Y and neuropeptide S) involved in the brain-gut connection by altering the activity of protease-activated receptor-2 (PAR-2). PAR-2 influences mood, inflammation, immune response, perception of injury, and allergic reactions. It not only increases intestinal permeability but also enhances the sensitivity of gastrointestinal organs in response to psychological stress [22]. Moreover, activation of MCs in colon mucosa induces abdominal pain by interacting with enteric nerves and disrupting tight junction molecules in the intestinal epithelium. In addition, it compromises the integrity of the intestinal epithelial barrier by triggering the release of MCT [23]. Approximately 70% of MCs in the intestinal lining are highly innervated. When stimulated by spinal nerves, these MCs release various mediators, including histamine, serotonin, MCT, and IL-6. These mediators subsequently activate neurons in the submucosal layer of the intestines. These signals are eventually transmitted to the central nervous system, where they are interpreted as pain [24]. Finally, the release of 5-HT by intestinal cells can stimulate MCs, resulting in localized abnormal changes within the intestines. This disruption can affect intestinal movement, blood flow, and secretion, ultimately impacting intestinal transit function and the composition of intestinal fluids, potentially leading to IBS [25]. SYNC can not only regulate the brain-gut axis by soothing the liver and invigorating the spleen but also diminish the infiltration of MCs in the colon, thereby modulating intestinal function.

Diverse signals, including stress, can activate the Nr4a nuclear receptor subfamily, also known as the orphan receptor. The Nr4a nuclear family has three members: Nur77/Nr4a1, Nurr1/Nr4a2, and NOR-1/Nr4a3. These members are considered early-response genes that respond to various stimuli such as cytokines, infectious agents, growth factors, and cellular stress. They improve tolerance in peripheral T cells in response to chronic antigenic stimuli and also prevent improper B-cell responses to self-antigens [26]. Nr4a3 is a transcription factor involved in multiple functions in vascular biology [27, 28], inflammation [29, 30], glycolipid metabolism [31] and tumor immunity [32]. This study found that the gene Nr4a3 was significantly increased following PI3K/AKT or PKC/ERK pathways activation during MCs activation. Nr4a3 was involved in the control of cytokines and chemokines release during MCs activation by interacting with the high-affinity IgE receptor, FceRI. This receptor mediate cytokine production during MCs activation, indicating that Nr4a3 can regulate the MC function [33–35]. In addition, activated CRH triggers the production of Nr4a3 by MCs thereby modulating the brain-gut communication pathway known as the HPA axis. This pathway coordinates stress response, intestinal peristalsis, permeability, and inflammatory response. These mechanisms are associated with occurrence of mood disorders, increased visceral sensitivity, and fluctuations in colonic integrity in individuals with IBS [10, 36]. A previous study reported that the Nr4a3 gene was upregulated in MCs located near CGRP-immunoreactive fibers in the colon of WAS mice. However, the CGRP-induced activation of MCs was decreased when

Nr4a3 was silenced [33]. Nr4a3 has also been influence mood changes. A meta-analysis found that individuals with IBS were three times more prone to experience anxiety or depression compared to healthy individuals [37]. Animal studies demonstrated showed that rats injected multiple times with Nr4a3 shRNA through the lateral ventricle exhibited significant improvement in depressive symptoms. Our RNA-Seq analysis and *in vivo* experiments revealed that the Nr4a3 protein and gene were unregulated in the model group. SYNC treatment decreased Nr4a3 expression, as well as the expression of 5HT and CRH-R1, thereby improved the mood of IBS-D rats. This suggests a potential link between Nr4a3 and the regulation of the HPA axis. Therefore, Nr4a3 and its associated genes are potential targets for developing treatments that suppress visceral sensitivity and alleviate depression in IBS-D [38].

SYNC is an effective treatment for IBS-D, derived from the modification of classic prescriptions. In this study, we found that SYNC treatment decreased the fecal water content and AWR score, regulated MCs activation, inhibited MCs degranulation, reduced MCT expression, and improved symptoms in IBS-D rats. To explore the mechanism of action of SYNC in the treatment of IBS-D at the genetic level and identify potential signaling pathways, we performed transcriptomic analysis of rat colon tissues. In the SYNC, IBS-D, and PI groups, Nr4a3 and PI3K were identified as differentially expressed genes. The expression of these genes was lowest in the control group and highest in the model group. However, their expression levels were decreased following drug treatment. GO enrichment analysis revealed that SYNC could treat IBS-D mainly by regulating cell activation, the interaction of cytokine receptors, and intestinal immune regulation. KEGG analysis further showed that the Nr4a3 and PI3K pathways were activated, suggesting that SYNC may inhibit MCs degranulation and regulate the intestinal immune environment through the Nr4a3/PI3K signaling pathway. Research has demonstrated that the interaction between intestinal immune cells and sensory nerves may influence the occurrence of IBS. Increased activation of intestinal mucosal immunity has also been observed in IBS patients, providing new directions for further research on IBS [39].

IL-6 is a common pro-inflammatory factor in the intestines. Analysis of blood samples from the model group revealed that IL-6 levels were elevated, indicating the presence of intestinal inflammation. SYNC treatment significantly reduced IL-6 levels. Moreover, CGRP, acting through its receptor complex in the bed nucleus of the stria terminalis (BNST), interacted with CRH and activated MCs through a CRH-R1-dependent mechanism, thereby contributing to anxiety. Western blot results revealed an upregulation of CRH-R1 expression in both the colon and hypothalamus of IBS-D rats. ELISA results demonstrated significantly elevated levels of serum CRH and CGRP expression in the model group, accompanied by an increased number of activated MCs, consistent with the findings of colonic toluidine blue staining. Activation of MCs releases various cytokines and biologically active substances, which participate in the generation of immune inflammatory response. MCT, a specific marker produced after MCs activation, was significantly higher in the colon of rats in the model group than in the other groups, and it was decreased in all groups to varying degrees after intervention, suggesting that SYNC may improve IBS-D symptoms by inhibiting MCs activation.

Research has shown that Tetrahydropalmatine [40, 41] and Hesperetin can inhibit inflammation and oxidative stress via the NF-κB or PI3K/AKT signaling pathways [42]. Paeoniflorin can regulate mood by targeting the HPA axis and the levels of neurotransmitters. It exerts anti-inflammatory and immunomodulatory effects by inhibiting the degranulation of MCs [43, 44]. Atractylenolide III inhibits hippocampal neuronal inflammation triggering anxiolytic-depressive effects [45], and also downregulates histamine and IL-6 levels following MCs activation to modulate immune response [46]. Phenolic acids like Ferulic acid regulate NF-κB associated pathways to induce anti-inflammatory and anti-apoptotic effects in rats with enteritis [47]. Gallic acid can prevent the formation of colonic adhesins and cuprocytes, thereby

improving intestinal inflammation and reducing oxidative stress [48]. Our findings suggest that the chemical components of SYNC can regulate inflammation and immune stress in the intestines. Different Chinese herbs exert therapeutic effects on IBS and can be adjusted based on symptoms [49]. However, further research is needed to understand its role in mast cell degranulation and whether its effects are mediated through the Nr4a3-related pathway. Pathways form a network, and there are interconnections between different pathways. KEGG and GO enrichment analysis also identified additional pathways potentially associated with SYNC's treatment of IBS, such as the IL-17 signaling pathway, NF-κB signaling pathway, TNF signaling pathway, NOD-like receptor signaling pathway, and MAPK signaling pathway, all of which are associated with the alleviation of low-grade inflammation and reduction of symptoms associated with IBS by SYNC. In addition, NF-κB and MAPK signaling pathways are involved in the regulation of depression. Cytokine stress can induce depression and also promote inflammatory responses by regulating sympathetic and parasympathetic nervous system pathways. Depressive symptoms may arise as a behavioral consequence of the evolutionary advantage provided by genes promoting inflammation. Targeting pro-inflammatory cytokines and their signaling pathways presents a novel approach to treating depression. Building on animal experiments, we propose that SYNC can alleviate depression by downregulating the expression of CRH-R1 and Nr4a3 in the hypothalamus. In addition, it may reduce visceral hypersensitivity by inhibiting mast cell activation through the decreased phosphorylation of Nr4a3 and PI3K.

Safety is the most important consideration during the design of pharmaceuticals. The complex and specific nature of traditional Chinese medicine has raised concerns about its safety. In response, China established an adverse drug reaction monitoring system in 1999. According to the *Annual Report on National Adverse Drug Reaction Detection*, Chinese medicine, excluding injections, generally exhibit favorable safety. In our study, the SYNC formula was orally administered, which maximizes its safety. None of the drugs in SYNC fall under categories with toxic side effects, such as "Eighteen incompatible herbs and Nineteen herbs of mutual antagonism." Furthermore, SYNC did not show any notable toxic side effects in clinical practice. However, these observations do not imply absolute safety, and we aim to explore its long-term safety to prevent potential toxic side effects. Moreover, considering the personalized nature of TCM prescriptions, clinical treatment should be tailored according to individual symptoms.

## Conclusion

In summary, SYNC can improve visceral hypersensitivity and diarrhea in rats with IBS-D by regulating the activation of intestinal mucosal MCs. The key gene involved in this activation is Nr4a3. This therapeutic process is closely linked to immune response, inflammatory pathways, and cell adhesion molecules in the intestines.

## Supporting information

**S1 Fig.**
(TIF)

**S1 Data.**
(XLSX)

**S1 Raw images.**
(PDF)

## Acknowledgments

We thank the members of the Basic Laboratory of Chongqing Hospital of Traditional Chinese Medicine for providing support for this study.

## Author Contributions

**Conceptualization:** Xiao-Jun Yang.

**Data curation:** Yajing Guo.

**Formal analysis:** Yajing Guo.

**Funding acquisition:** Qiongqiong Lu, Xiao-Jun Yang.

**Investigation:** Xiao-Jun Yang.

**Methodology:** Yajing Guo, Yuxi He, Min Duan, Nvping Liu, Xin Wu.

**Project administration:** Yue Wu, Min Duan, Nvping Liu, Xin Wu, Yuanjun He.

**Resources:** Yue Wu, Ting Li.

**Software:** Ting Li.

**Supervision:** Qiongqiong Lu, Xiao-Jun Yang, Baijun Qin.

**Writing – original draft:** Yajing Guo.

**Writing – review & editing:** Yajing Guo.

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
