## [Decision Letter · Decision Letter 0]

8 Dec 2023

PONE-D-23-36123Exploring the mechanism of Shu-yi-ning-chang decoction in modulating the Nr4a3 pathway to ameliorate visceral hypersensitivity in irritable bowel syndromePLOS ONE

Dear Dr. yang,

Thank you for submitting your manuscript to PLOS ONE. After careful consideration, we feel that it has merit but does not fully meet PLOS ONE’s publication criteria as it currently stands. Therefore, we invite you to submit a revised version of the manuscript that addresses the points raised during the review process.

We look forward to receiving your revised manuscript.

Kind regards,

Xuan Zeng

Academic Editor

PLOS ONE

Comments from Senior Staff Editor: We note that one or more reviewers has recommended that you cite specific previously published works. As always, we recommend that you please review and evaluate the requested works to determine whether they are relevant and should be cited. It is not a requirement to cite these works. We appreciate your attention to this request.

3. To comply with PLOS ONE submissions requirements, in your Methods section, please provide additional information regarding the experiments involving animals and ensure you have included details on (1) methods of sacrifice, (2) methods of anesthesia and/or analgesia, and (3) efforts to alleviate suffering.

“Chongqing Natural Science Foundation (No: cstc2021jcyj-msxmX0858) and National Natural Science Foundation of China (82205085).”

6. PLOS requires an ORCID iD for the corresponding author in Editorial Manager on papers submitted after December 6th, 2016. Please ensure that you have an ORCID iD and that it is validated in Editorial Manager. To do this, go to ‘Update my Information’ (in the upper left-hand corner of the main menu), and click on the Fetch/Validate link next to the ORCID field. This will take you to the ORCID site and allow you to create a new iD or authenticate a pre-existing iD in Editorial Manager. Please see the following video for instructions on linking an ORCID iD to your Editorial Manager account: https://www.youtube.com/watch?v=_xcclfuvtxQ.

7. PLOS ONE now requires that authors provide the original uncropped and unadjusted images underlying all blot or gel results reported in a submission’s figures or Supporting Information files. This policy and the journal’s other requirements for blot/gel reporting and figure preparation are described in detail at https://journals.plos.org/plosone/s/figures#loc-blot-and-gel-reporting-requirements and https://journals.plos.org/plosone/s/figures#loc-preparing-figures-from-image-files. When you submit your revised manuscript, please ensure that your figures adhere fully to these guidelines and provide the original underlying images for all blot or gel data reported in your submission. See the following link for instructions on providing the original image data: https://journals.plos.org/plosone/s/figures#loc-original-images-for-blots-and-gels.

8. We notice that your supplementary tables are included in the manuscript file. Please remove them and upload them with the file type 'Supporting Information'. Please ensure that each Supporting Information file has a legend listed in the manuscript after the references list.

Reviewers' comments:

Reviewer's Responses to Questions

**Comments to the Author**

1. Is the manuscript technically sound, and do the data support the conclusions?

Reviewer #1: Yes

Reviewer #2: Partly

2. Has the statistical analysis been performed appropriately and rigorously? 

Reviewer #1: No

Reviewer #2: I Don't Know

3. Have the authors made all data underlying the findings in their manuscript fully available?

Reviewer #1: No

Reviewer #2: Yes

4. Is the manuscript presented in an intelligible fashion and written in standard English?

Reviewer #1: No

Reviewer #2: No

5. Review Comments to the Author

Reviewer #1: 1. The study title can be more concise while maintaining clarity. For instance, "Efficacy of Shu-yi-ning-chang Decoction on IBS-D: Modulating Nr4a3 Pathway to Reduce Visceral Hypersensitivity" might be more succinct.

2. "The clinical manifestations of IBS primarily include abdominal pain, irregular bowel movements ..." - the definition of IBS cited is incomplete. It is a combination of chronic abdominal pain associated with a change in the frequency or form of stool.

3. "The precise pathogenesis of IBS remains elusive, though it is generally associated with several mechanisms, including visceral hypersensitivity, dysregulation of the gut microbiota, increased intestinal epithelial permeability, brain-gut interaction disorder, eating disorders and epigenetic alterations" - I am not sure where the 'eating disorders' come from. In addition to ref [3,4], suggest to mention that the pathophysiology of IBS is currently thought to represent a complex interplay among the gut microbiota, mucosal immune system, impaired mucosal barrier function, visceral hypersensitivity, gut motility, and alterations in the gut-brain axis (citation: ncbi.nlm.nih.gov/pmc/articles/PMC6159811).

4. "However, these methods are typically palliative, marked by limitations and high recurrence rates" - 'palliative' is the wrong word here. I think you mean 'symptomatic relief'.

5. "... dispersing stagnant liver qi to alleviate qi stagnation, strengthening the spleen to solidify the root and cultivate the elements, fostering communication between the brain and intestines" - I am not sure what "solidify the root" means. It may be helpful to have a schematic here to explain this.

6. Curcumin is also a common ingredient found in TCM concoctions, suggest to cite a previous study here to illustrate the use of herbal therapies for irritable bowel syndrome (IBS) symptoms (citation: ncbi.nlm.nih.gov/pmc/articles/PMC6210149).

7. The specific focus on IBS-D should be explained at least briefly in the introduction section.

8. Scientific names such as "Radix Angelicae Sinensis" should be written as "Angelica sinensis radix" and in italics.

9. More details on the steps taken to minimize pain and discomfort in the animal subjects would be beneficial, addressing ethical considerations.

10. The nature of replication in the experimental design is unclear, and the assessment of uncertainty in the reported measurement is absent or unclear. Information on the number of replicates conducted for each experiment and the variation observed would add credibility to the findings.

11. Please state the exact p values and their associated 95% CI wherever possible. Simply stating P < 0.05 is neither sufficient nor informative.

12. The study seems to imply causation (SYNC affecting the Nr4a3 pathway), but it is crucial to distinguish this from mere correlation, especially in a complex system involving multiple pathways. Moreover, the results from an animal model may not be directly applicable to humans. The limitations in translating these findings to human patients should be discussed.

13. The discussion should consider and address alternative explanations for the findings. This includes the possibility of other pathways being affected by SYNC.

14. The original figures for the western blots should be supplied for review. Some of the western blot bands were over-loaded or over-exposed, and they were not quantified or statistically analyzed.

15. For Figures 3 and 5, please provide the scale bars and also state explicitly in the legend the stain used, e.g. "Amyloid is birefringent after Congo red staining (viewed with polarized light)".

Reviewer #2: This manuscript investigated the effects and the underlying mechanism of SYNC on IBS-D via pharmacodynamic evaluation, RNA-Seq and WB, which is helpful to reveal the modern application principle of SYNC. However, there are still quite a few questions in this paper that need to be revised and improved. Unless these issues are addressed, this paper cannot be considered for publication.

1. What is the role of the part “Identified chemical components in SYNC” in the article? How does it relate to the preceding and following contents? If this section is not required, suggest deleting it.

2. It is recommended that the results of the determination of the content of the main components of SYNC be provided as a basis for quality control of SNYC.

3. In the method section “Animals”, the authors wrote about a total of 12 pregnant rats. However, in the “Induction of IBS-D rat model and experimental design”, the authors wrote “Pregnant rats were randomly assigned to one of two groups: the control group (n=8), which received no interventions and had a normal diet and access to water, and the model group (n=50)”, it's confusing. It is recommended to write clearly in the animal experiment design section that offspring male rats were used, and how many in total. The presentation should be clear.

4. The description of the pathological images is not clear. It is recommended to make revisions to improve the readability of the manuscript. For example, Clearly describe which group the object is, and compare the pathological changes with which group?

5. Please label the lesion location in pathological images.

6. The authors are advised to label the names of significant genes in the volcano plot of Fig. 7B, D and F.

7. RNA-seq results have identified numerous differential genes, why focus on Nr4a3 and PI3K in particular? Are Nr4a3, PI3K, AKT, and CRH-R1 related to the pathway depicted in Fig.7M? It is recommended to draw a schematic diagram of the signaling pathway and explain it.

8. Of the pathways/genes affected by SYNC, which are related to alleviating IBS and which are related to antidepressant effects? Based on the experiments, do the authors believe that the primary role of SYNC is to act as an antidepressant or to alleviate IBS?

6. PLOS authors have the option to publish the peer review history of their article (what does this mean?). If published, this will include your full peer review and any attached files.

Reviewer #1: No

Reviewer #2: No

---

## [Author Response · Author response to Decision Letter 0]

21 Dec 2023

Journal requirements

Thank you for your advice. The revised manuscript has been ensured to meet PLOS ONE's style requirements. Animal ethics has also been added, all the data used for the drawing has been provided as Supporting Information to be uploaded into the system, ORCID information has been submitted in the website, and the original images about WB have been uploaded.

Reviewer #1

We appreciate your kind suggestions and we have amended the manuscript accordingly. In addition, the followings are answers for your questions:

1. Thank you for pointing out this problem in manuscript, the issue regarding the title of the article has been corrected (Efficacy of Shu-yi-ning-chang Decoction on IBS-D: Modulating Nr4a3 Pathway to Reduce Visceral Hypersensitivity).

2. We agree with this comment and rewrote this portion of the revised draft as follows: “IBS is mainly characterized by persistent or intermittent abdominal pain and bloating, which is related to changes in the frequency or form of stool. It also includes a range of physical, visceral, or psychological comorbidities”

3. As suggested by the reviewer, we have deleted "eating disorders" and cited the relevant paper (5.Ng, Q.X., Soh, A.Y.S., Loke, W., Lim, D.Y., Yeo, W.S., 2018. The role of inflammation in irritable bowel syndrome (IBS). J. Inflamm. Res. 11, 345-349. https://doi.org/10.2147/JIR.S174982).

4.As the reviewer understands, the word "palliative" does mean "symptomatic" and we have corrected it: these methods usually can only temporarily relieve symptoms, marked by limitations and high recurrence rates.

5.Based on the reviewer's question, we have rephrased "solidify the root" to make its meaning clearer: strengthening the spleen can enhance the body's resistance.

6.Thank you very much for your suggestion, we found this article very valuable and have made it a reference (49.Ng, Q.X., et al., A Meta-Analysis of the Clinical Use of Curcumin for Irritable Bowel Syndrome (IBS). J Clin Med, 2018. 7(10).)

7.Thanks to your valuable suggestions, we have added content about IBS in the introduction section on page 4.

8.Thanks to your careful review, we have modified the writing of the names of the herbs, such as Radix bupleuri (Chai Hu), Angelica sinensis radix (Dang Gui),

9.We apologize for our oversight, content on animal ethics has been added, such as the use of sodium pentobarbital (40 mg/kg, I.p) for pain relief.

10.We appreciate your feedback, we have accordingly increased the number of relevant experimental repetitions to make the experimental design clearer and more plausible.

11.We appreciate your suggestion, we have stated the exact p-values and 95% CI as much as possible in the manuscript, there is no exact p-value when p < 0.001, so it is written as p < 0.001. It is difficult to list the exact p-values for each group when there is a uniform comparison between multiple groups, so it is written as uniform p-value.

12.Thank you for pointing this out, the causal relationship between SYNC and nr4a3 does need to be further verified, we are also doing in vitro silencing and overexpression experiments of the relevant pathway using RBL-2H3 cells, and we hope to continue to collaborate with your journal. SYNC is an effective prescription for the clinical treatment of IBS-D, and clinical trials on it are ongoing. According with your advice, we have added the limitations section in the discussion section accordingly on page 25.

13.Considering the reviewer's suggestion, we address alternative pathways (IL-17 signaling pathway, NF-κB signaling pathway, TNF signaling pathway, NOD-like receptor signaling pathway and MAPK signaling pathway) related to SYNC treatment of IBS on page 24.

14.We have provided the original figures for the western blots and performed quantitative statistical analyses.

15.Following the reviewers' suggestions, we have provided a scale of Figures 3 and 5 (400×) in the revised manuscript and labeled the stains used (MCs are blue-purple after toluidine blue staining).

Reviewer #2

Thank you for your nice comments on our article. According to your suggestions, we have supplemented several data here and corrected several mistakes in our previous draft. 

1.This section on "Identified chemical components in SYNC" is included due to the complexity of Chinese medicine compound prescriptions, which consider multiple symptoms. By utilizing HPLC, we can gain a deeper understanding of the prescription's function and gather insights for further research on its primary chemical components.

2.Thank you very much for your valuable comments. Currently, we have only analyzed the primary chemical components of SYNC based on the prescription and HPLC. This analysis was aimed at understanding the composition of the prescription. Following your suggestion, our team will focus on examining the content of SYNC in our future work.

3.We apologize for the unclear presentation and we have corrected it in the revised manuscript. There were a total of 12 pregnant rats in the animal experiments, and group experiments using offspring rats were conducted with 8 rats in each group, totaling 48 rats.

4.Thanks to your careful review, we have re-evaluated the colon pathology images to enhance the comparison between each groups and improve the clarity of the results analysis. (Compared with the normal group, the colonic mucosal structure in the model group remained largely intact, with widened intestinal glandular spaces. Numerous specimens exhibited substantial inflammatory cell infiltration and submucosal edema. In contrast to the model group, the colonic mucosa in the PI and SYNC-H groups demonstrated structural integrity, with glands arranged regularly, distinct layer structure, and a normal number of cup cells. Some inflammatory cell infiltration was observed. Relative to the SYNC-H group, the SYNC-L and SYNC-M groups exhibited more pronounced inflammatory cell infiltration in the colonic mucosa and wider gaps between intestinal glands).

5.Thank you for your suggestion, we have marked the location of the lesion in the pathological images: Black arrows points to lymphocytic infiltration, red arrows points to eosinophilic infiltration, and blue arrows indicate widening of the intestinal glandular spaces in comparison with the normal group, suggesting changes such as edema when the sample is fresh.

6.We are sorry that it is difficult to mark the exact location of the differential genes in the volcano map in Fig. 7 due to the ponderous number of genes clustered in the gray area. 

7.We fully understand the concerns of the reviewers, the previous literature search revealed a connection between the Nr4a family and IBS. Specifically, Nr4a3 was found to not only influence mood through the brain-gut axis, but also to interact with mast cells and play a role in the immune-inflammatory process, which are associated with IBS. Consequently, Nr4a3 and PI3K were chosen as the primary genes of interest. We mapped the relevant pathways as follows.

8.Thank you for pointing out this problem. Based on the transcriptome results, many of the pathways impacted by SYNC are linked to immune inflammation, including the IL-17, NF-κB, TNF, NOD-like receptor and MAPK signaling pathways. This suggests that SYNC may have potential in reducing low-grade inflammation in the intestines. In the Nr4a3/PI3K pathway, which is our main focus, the Nr4a3 gene not only regulates mood but also has connections to immune inflammation. SYNC is a TCM prescription based on the holistic concept, regulating the liver qi to improves negative emotions, regulating the spleen and stomach to strengthen the gastrointestinal function, which is a comprehensive prescription.

---

## [Decision Letter · Decision Letter 1]

25 Jan 2024

PONE-D-23-36123R1Efficacy of Shu-yi-ning-chang decoction on IBS-D: modulating Nr4a3 pathway to reduce visceral hypersensitivityPLOS ONE

Dear Dr. yang,

Thank you for submitting your manuscript to PLOS ONE. After careful consideration, we feel that it has merit but does not fully meet PLOS ONE’s publication criteria as it currently stands. Therefore, we invite you to submit a revised version of the manuscript that addresses the points raised during the review process.

We look forward to receiving your revised manuscript.

Kind regards,

Xuan Zeng

Academic Editor

PLOS ONE

Additional Editor Comments:

The author should think about how to correct the paper's shortcomings, not avoid the problem.

Reviewers' comments:

Reviewer's Responses to Questions

**Comments to the Author**

1. If the authors have adequately addressed your comments raised in a previous round of review and you feel that this manuscript is now acceptable for publication, you may indicate that here to bypass the “Comments to the Author” section, enter your conflict of interest statement in the “Confidential to Editor” section, and submit your "Accept" recommendation.

Reviewer #1: (No Response)

Reviewer #2: (No Response)

2. Is the manuscript technically sound, and do the data support the conclusions?

Reviewer #1: Partly

Reviewer #2: Partly

3. Has the statistical analysis been performed appropriately and rigorously? 

Reviewer #1: Yes

Reviewer #2: I Don't Know

4. Have the authors made all data underlying the findings in their manuscript fully available?

Reviewer #1: Yes

Reviewer #2: (No Response)

5. Is the manuscript presented in an intelligible fashion and written in standard English?

Reviewer #1: No

Reviewer #2: No

6. Review Comments to the Author

Reviewer #1: 1. The manuscript is still in general need of wordsmithing, several instances of awkward phrasing and grammatical lapses are present.

2. "... regulating the liver qi to improves negative emotions, regulating the spleen and stomach to strengthen the gastrointestinal function, which is a comprehensive prescription" - what exactly does this mean?

3. The manuscript could benefit from more detailed methodological descriptions to ensure reproducibility. Specifics on dosages, administration methods, and criteria for evaluation would be helpful. The use of UPLC-MS/MS for analyzing SYNC components should be elaborated, including specifics about the standards and calibration used.

4. The safety profile of SYNC, especially when considering long-term use, is not discussed. Herbal formulations can have variable compositions, and safety cannot be assumed.

Reviewer #2: 1. What is the role of the part “Identified chemical components in SYNC” in the article? How does it relate to the preceding and following contents? This section is not required, suggest deleting it.

2. The manuscript has a revised format that has not been processed, it may not be allowed to be submitted to the plos one system.

3. RNA-seq results have identified numerous differential genes, why focus on Nr4a3 and PI3K in particular? Are Nr4a3, PI3K, AKT, and CRH-R1 related to the pathway depicted in Fig.7M? It is recommended to draw a schematic diagram of the signaling pathway and explain it. I couldn't find the figure that added by the author.

4. Of the pathways/genes affected by SYNC, which are related to alleviating IBS and which are related to antidepressant effects? Based on the experiments, do the authors believe that the primary role of SYNC is to act as an antidepressant or to alleviate IBS? I asked this question in the hope that the authors would consider it and supplement the RNA-seq analysis in the manuscript.

7. PLOS authors have the option to publish the peer review history of their article (what does this mean?). If published, this will include your full peer review and any attached files.

Reviewer #1: No

Reviewer #2: No

---

## [Author Response · Author response to Decision Letter 1]

2 Feb 2024

Dear Reviewers, 

Thank you very much for your professional advice. We have carefully revised the manuscript by incorporating all the suggestions by the review panel. Once again, We have invited native English speakers to touch up the English grammar and coherence of the article (Manuscript Number: PONE-D-23-36123). At the end of the letter, we have attached the English editing certificate.

The following is a summary list of changes: 

Reviewer #1

We appreciate your further professional review work on our article. As you are concerned, there are several problems that need to be addressed. According to your nice suggestions, we have made extensive corrections to our previous manuscript. We have added necessary sections to the manuscript to complete the article. The detailed corrections are listed below.

1. We apologize for our oversight and have reworked the grammar and phrasing of the article.

2. "... regulating the liver qi to improves negative emotions, regulating the spleen and stomach to strengthen the gastrointestinal function, which is a comprehensive prescription" is a TCM term in Chinese medicine theory, meaning that regulating the qi of the liver and the function of the spleen and stomach can achieve the effect of improving depression and strengthening the function of the intestinal tract, and this method is extremely effective in treating IBS.

3. Thank you very much for your very useful suggestions, as we have elaborated on the specific dosages, methods of administration and evaluation criteria in the manuscript. The dosage of each Chinese medicine is supplemented in detail in the drug component. The section "Identified chemical components in SYNC" has been deleted at the suggestion of reviewer 2.

Specifics on dosages: Radix bupleuri (Chai Hu) 15g, Angelica sinensis radix (Dang Gui) 10g, Fructus Aurantii (Zhi Qiao) 10g, Radix Paeoniae Alba (Bai Shao) 10g, Rhizoma Corydalis(Yuan Hu) 10g, Radix Pseudostellariae (Tai Zi Shen) 15g, Poria (Fu Ling) 10g, Rhizoma Atractylodis Macrocephalae (Bai Zhu) 10g, Dried tangerine peel (Chen Pi) 10g, Pericarpium Granati (Shi Liu Pi) 10g, Bergamot (Fo Shou) 10g, Citron (Xiang Yuan) 10g.

Dosages and methods of administration: Successfully modeled rats were randomly assigned to one of five groups, the model group, SYNC low-dose (6.825 g/kg/d), medium-dose (13.65 g/kg/d), high-dose (27.3 g/kg/d), and Pinaverium Bromide group (15.75 mg/kg/d). The drug dosages were adjusted based on equivalent dosages for adults and rats and administered via gavage for 14 days.

Criteria for evaluation: 1. Fecal water content. Fecal water content (%) was determined by collecting feces from each rat before and after molding between 8:00 and 9:00. The collected feces were weighed and recorded as wet weight. Next, they were dried in an oven at 120℃ for 30 min and weighed again, with this recorded as the dry weight. This process was repeated until the weight of the feces no longer changed. Fecal water content was calculated as follows: feces water content (%) = (wet weight of feces+dry weight of feces)/ wet weight of feces×100% [13]). 2. Body weight:Additionally, rat body weights were recorded every five days, starting from the end of the modeling period (week 0) and continuing until the conclusion of the experiment, totaling six measurements. 3. Abdominal withdrawal reflex (AWR) scoring: AWR scores were assigned to rats before and after treatment. Rats were fasted but provided with water 24 h before measurement. A catheter lubricated with paraffin oil was gently inserted into the rat’s anus and secured to the tail with medical tape to prevent dislodgment. The rats were then placed in a transparent container, and room-temperature pure water was injected into a balloon through the catheter. Each injection lasted for 20 s, and the water volume injected was noted when the AWR score was 3. This process was repeated 3 times with a 15 min interval between injections, and the average value was used to determine the minimum volume threshold for AWR caused by non-injurious colorectal dilatation in these rats. 

4. Thanks to your valuable suggestions, we added a section on drug safety to the discussion of the manuscript.

 (Safety is the most important consideration during the design of pharmaceuticals. The complex and specific nature of traditional Chinese medicine has raised concerns about its safety. In response, China established an adverse drug reaction monitoring system in 1999. According to the Annual Report on National Adverse Drug Reaction Detection, Chinese medicine, excluding injections, generally exhibit favorable safety. In our study, the SYNC formula was orally administered, which maximizes its safety. None of the drugs in SYNC fall under categories with toxic side effects, such as "Eighteen incompatible herbs and Nineteen herbs of mutual antagonism." Furthermore, SYNC did not show any notable toxic side effects in clinical practice. However, these observations do not imply absolute safety, and we aim to explore its long-term safety to prevent potential toxic side effects. Moreover, considering the personalized nature of TCM prescriptions, clinical treatment should be tailored according to individual symptoms).

Reviewer #2

 We appreciate your detailed and constructive comments. According to your suggestions, we have supplemented relevant content here and corrected several mistakes in our previous manuscript. We deleted some unneeded content and explained and revised the manuscript according to each suggestion one by one. The detailed corrections are listed below.

1. Thank you very much for your suggestion. After careful consideration, we have deleted the section "Identified chemical components in SYNC".

2. We apologize for our oversight and have carefully revised the formatting of the manuscript to ensure compliance with PLOS ONE.

3. Thanks to your proposal, we have mapped the relevant access roads as follows: 

(To prevent the image from being lost again, name the image in the article submission system: Signaling Pathway)

4. We appreciate your valuable feedback, and we have included an analysis of the pathways and genes impacted by SYNC in the manuscript： 

(KEGG and GO enrichment analysis also identified additional pathways potentially associated with SYNC's treatment of IBS, such as the IL-17 signaling pathway, NF-κB signaling pathway, TNF signaling pathway, NOD-like receptor signaling pathway, and MAPK signaling pathway, all of which are associated with the alleviation of low-grade inflammation and reduction of symptoms associated with IBS by SYNC. In addition, NF-κB and MAPK signaling pathways are involved in the regulation of depression. Cytokine stress can induce depression and also promote inflammatory responses by regulating sympathetic and parasympathetic nervous system pathways. Depressive symptoms may arise as a behavioral consequence of the evolutionary advantage provided by genes promoting inflammation. Targeting pro-inflammatory cytokines and their signaling pathways presents a novel approach to treating depression. Building on animal experiments, we propose that SYNC can alleviate depression by downregulating the expression of CRF-R1 and Nr4a3 in the hypothalamus. In addition, it may reduce visceral hypersensitivity by inhibiting mast cell activation through the decreased phosphorylation of Nr4a3 and PI3K).

Once again, thank you very much for your suggestions. And we hope that the revised manuscript can be accepted by PLOS ONE. If further revision is necessary, please contact me.

Thank you and best regards.

Sincerely yours,

Ya-Jing Guo

Corresponding author:

Name: Xiao-Jun Yang

---

## [Editor Report · Decision Letter 2]

9 Feb 2024

Efficacy of Shu-yi-ning-chang decoction on IBS-D: modulating Nr4a3 pathway to reduce visceral hypersensitivity

PONE-D-23-36123R2

Dear Dr. yang,

We’re pleased to inform you that your manuscript has been judged scientifically suitable for publication and will be formally accepted for publication once it meets all outstanding technical requirements.

Kind regards,

Xuan Zeng

Academic Editor

PLOS ONE
---

## [Editor Report · Acceptance letter]

27 Feb 2024

PONE-D-23-36123R2 

PLOS ONE

Dear Dr. Yang, 

I'm pleased to inform you that your manuscript has been deemed suitable for publication in PLOS ONE. Congratulations! Your manuscript is now being handed over to our production team.

Kind regards, 

on behalf of

Dr. Xuan Zeng 

Academic Editor

PLOS ONE